# Blood Cadmium Levels and Oxygen Desaturation during the 6-Minute Walk Test in Patients with Chronic Obstructive Pulmonary Disease

**DOI:** 10.3390/medicina57111160

**Published:** 2021-10-25

**Authors:** Li-Chung Chiu, Ping-Chih Hsu, Tzung-Hai Yen, Scott Chih-Hsi Kuo, Yueh-Fu Fang, Yu-Lun Lo, Shu-Min Lin, Cheng-Ta Yang, Chung-Shu Lee

**Affiliations:** 1Department of Thoracic Medicine, Chang Gung Memorial Hospital, College of Medicine, Chang Gung University, Taoyuan 33305, Taiwan; pomd54@cgmh.org.tw (L.-C.C.); 8902049@gmail.com (P.-C.H.); chihhsikuo@gmail.com (S.C.-H.K.); dr.fang.yf@gmail.com (Y.-F.F.); loyulun@hotmail.com (Y.-L.L.); smlin100@gmail.com (S.-M.L.); yang1946@cgmh.org.tw (C.-T.Y.); 2Graduate Institute of Clinical Medical Sciences, College of Medicine, Chang Gung University, Taoyuan 33302, Taiwan; 3Department of Nephrology, Chang Gung Memorial Hospital, College of Medicine, Chang Gung University, Taoyuan 33305, Taiwan; m19570@cgmh.org.tw; 4Clinical Poison Center, Kidney Research Center, Center for Tissue Engineering, Chang Gung Memorial Hospital, Taoyuan 33305, Taiwan; 5Department of Internal Medicine, Taoyuan Chang Gung Memorial Hospital, Taoyuan 33378, Taiwan; 6Department of Respiratory Therapy, College of Medicine, Chang Gung University, Taoyuan 33302, Taiwan; 7Department of Thoracic Medicine, New Taipei Municipal TuCheng Hospital, Chang Gung University, Taoyuan 33302, Taiwan

**Keywords:** cadmium, chronic obstructive pulmonary disease, 6-min walk test, desaturation

## Abstract

*Background and Objectives:* chronic obstructive pulmonary disease (COPD) is characterized by persistent airflow limitation and a history of exposure to noxious stimuli. Cigarette smoking is the most important causal factor for developing COPD. Cadmium, a minor metallic element, is one of the main inorganic components in tobacco smoke. Inhaled cadmium was associated with a decline in lung function, gas exchange impairment, and the development of obstructive lung disease. Patients with COPD who had oxygen desaturation during the 6-min walk test (6MWT) had a significantly worse prognosis than non-desaturation in COPD patients. Nonetheless, few studies have addressed the influence of blood cadmium levels on exercise-induced oxygen desaturation in COPD patients. Our objective was to assess the potential impact of blood cadmium levels on oxygen desaturation during the 6MWT among COPD patients. *Materials and Methods:* we performed a retrospective analysis of patients with COPD who were examined for blood cadmium levels in a tertiary care referral center in Taiwan, between March 2020 and May 2021. The 6-min walk test was performed. Normal control subjects who had no evidence of COPD were also enrolled. *Results:* a total of 73 COPD patients were analyzed and stratified into the high-blood cadmium group (13 patients) and low-blood cadmium group (60 patients). A total of 50 normal control subjects without a diagnosis of COPD were enrolled. The high-blood cadmium group had a significantly higher extent of desaturation than the low-blood cadmium group. The frequency of desaturation during 6MWT revealed a stepwise-increasing trend with an increase in blood cadmium levels. A multivariable logistic regression model revealed that blood cadmium levels were independently associated with desaturation during the 6MWT (odds ratio 12.849 [95% CI 1.168–141.329]; *p* = 0.037). *Conclusions:* our findings indicate that blood cadmium levels, within the normal range, were significantly associated with desaturation during 6MWT in patients with COPD.

## 1. Introduction

Chronic obstructive pulmonary disease (COPD) is characterized by persistent and progressive respiratory symptoms and airflow limitation due to airway and/or alveolar abnormalities and is usually caused by significant exposure to noxious particles or gases [1,2,3]. Tobacco smoking remains the major risk factor for COPD, but environmental exposures (biomass fuel exposure and air pollution) and host factors (genetic abnormalities, abnormal lung development, and accelerated aging) may be contributors [2].

Environmental exposure to heavy metals is linked to the pathogenesis of COPD, which could induce chronic inflammation and uncontrolled oxidative stress in the lungs, resulting in tissue destruction and manifesting as obstructive lung disease [4]. Cadmium, a toxic heavy metal, is a major component of cigarette smoke [4]. Cadmium exposure mainly occurs from smoking and through contaminated food and water intake or environmental exposure (occupational inhalation or air pollution). Inhaled cadmium is a known entity for increased risk of emphysema, COPD or lung cancer, deteriorated lung function, and gas exchange impairment [4,5,6].

The 6-min walk test (6MWT) is a practical, simple exercise test and a reproducible means to measure functional capacity in COPD patients; it provides information on walk distance and oxygen desaturation [7,8,9,10]. The 6-min walk distance (6MWD) and oxygen desaturation during the 6MWT have predictive values for clinical outcomes or mortality in patients with COPD [7,11,12,13].

Researchers have yet to investigate the association between blood cadmium levels and exercise-induced oxygen desaturation (EID) in patients with COPD. Therefore, the objective of this study was to determine the correlation between blood cadmium levels with desaturation during the 6MWT in patients with COPD.

## 2. Materials and Methods

### 2.1. Study Design and Patients’ Inclusion

This study was based on retrospective analysis of 173 patients with COPD aged 40–78 years from the outpatient clinic or hospital admissions between March 2020 and May 2021 at Chang Gung Memorial Hospital (CGMH) in Taiwan. Inclusion criteria were as follows: (1) age > 40 years (2) post-bronchodilator FEV_1_/FVC < 0.7 (3) both blood cadmium examination and 6MWT were performed. COPD patients who performed 6MWT were under the supervision of well-trained technicians at the pulmonary rehabilitation center in our institution. The 6MWT was performed in accordance with the standard protocol [8], and the 6MWT was done with room air, without supplementing the environment with oxygen.

Patients who had resting hypoxemia (defined by arterial oxygen saturation measured by pulse oximetry (SpO_2_) < 90% before 6MWT) or did not perform blood cadmium examination or 6MWT were excluded. Normal control subjects who had no evidence of COPD were also enrolled. The local Institutional Review Board for Human Research approved this study (CGMH IRB no. 202101368B0A3), and the need for informed consent was waived due to the retrospective nature of the study.

### 2.2. Definitions

COPD was diagnosed in accordance with the GOLD criteria with post-bronchodilator FEV_1_/FVC < 0.7 [1]. Oxygen desaturation during 6MWT or EID was defined as a fall in SpO_2_ of ≥4% between the end (post-test) and beginning (pre-test) and an end (post-test) SpO_2_ of <90% [9]. The distance-saturation product (DSP) is the product of the final walk distance in meters and the lowest oxygen saturation of the participants during 6MWT (as measured by pulse oximetry) [14].

### 2.3. Data Collection

Demographic data, underlying comorbidities, laboratory data, smoking status, acute exacerbations in the previous year, pulmonary function tests, and the results of 6MWT, such as SpO_2_, heart rate, and walk distance, were recorded.

### 2.4. Measurement of Blood Cadmium Levels

Blood cadmium measurements were conducted by inductively coupled plasma mass spectrometry (ICP-MS). Blood specimens were collected in 6 mL plastic blood collection tubes containing K2EDTA as an anticoagulant (BD, Franklin Lakes, NJ, USA). Blood specimens were stored at 4 °C. Cadmium was quantified by means of ICP-MS on a PerkinElmer NexION 350X instrument (Waltham, MA, USA), and analyzed using a no-gas mode. A 500 μL of blood specimens were diluted 10 times with 1.5% (*w*/*v*) nitric acid (JT Baker, Phillipsburg, NJ, USA) solution containing yttrium as an internal standard. The cadmium and yttrium standards were purchased from AccuStandard (New Haven, CT, USA). A standard calibration curve with a range of 0 to 40 μg/L was created. The calibration curve had a correlation coefficient of more than 0.995. Level 1 control in-house prepared control and level 2 control Seronorm trace elements whole blood control (Sero, Billingstad, Norway) were used and analyzed at the start and end of each analytical run and after every 10 samples. The lower limit of quantitation (LOQ) for cadmium by ICP-MS was 0.5 μg/L. Since the LOQ for cadmium was 0.5 μg/L, the values below 0.5 μg/L were assigned to LOQ (i.e., 0.5 μg/L) for analysis.

### 2.5. Statistical Analysis

Continuous variables were presented as mean and standard deviation for normally distributed variables or median and interquartile range for abnormally distributed variables. Student’s *t* test was performed to compare normally distributed data, and a Mann-Whitney *U* test was used for nonparametric data. Categorical variables were presented as frequencies and percentages and were compared by the chi-square test for equal proportions or the Fisher’s exact test. Receiver operating characteristic curve and Youden index were used to determine the cutoff to dichotomize continuous variables. Risk factors (independent variables) associated with desaturation during 6MWT (dependent variable) were analyzed using univariate analysis in the first step, followed by a multivariable logistic regression model with stepwise selection. Independent variables with a univariate value of *p* less than 0.20 were entered into the model of multiple logistic regression analysis. The results were presented using the odds ratio and 95% confidence interval (CI). All statistical analyses were performed with SPSS Statistics version 26.0. Statistical significance was considered when a two-sided *p-*value was less than 0.05.

## 3. Results

### 3.1. Study Patients

During the study period, a total of 91 patients with COPD examined for blood cadmium were included. After excluding 18 patients, a total of 73 patients with COPD were enrolled in the current analysis. A total of 16 patients (22%) who had desaturation during 6MWT (i.e., EID) were assigned to the desaturation group, and 57 patients (78%) to the no-desaturation group. The maximum Youden index value was used to categorize patients according to blood cadmium levels, using a cutoff point of 1.75 μg/L: high-blood cadmium group (13 patients; 18%) and low-blood cadmium group (60 patients; 82%) (Figure 1).

### 3.2. Comparisons of COPD Patients with or without Desaturation during the 6MWT

As shown in Table 1, the mean age was 70.6 ± 9.1 years, and almost all enrolled COPD patients had a history of smoking (91.8%, former or current). The values of blood cadmium were significantly higher in patients with COPD than in normal control subjects (1.29 ± 0.67 versus 1.03 ± 0.56, *p* = 0.024). There were no significant differences between the no-desaturation group and the desaturation group regarding age, gender, comorbidities, and smoking status. However, the desaturation group had significantly higher body weight, body mass index, hemoglobin levels, and blood cadmium levels than the no-desaturation group (all *p* < 0.05). The desaturation group had worse lung function tests (significantly lower FEV_1_ % predicted values and lower FEV_1_/FVC ratio), significantly lower pre-test and post-test saturation, and significantly lower DSP.

### 3.3. Comparing Patients with High- and Low-Blood Cadmium Levels

As shown in Table 2, no significant differences were observed between the high-blood-cadmium and low-blood cadmium groups regarding age, gender, body weight, body mass index, comorbidities, and laboratory data (except blood cadmium). Patients in the higher blood cadmium group had significantly lower FEV_1_ % predicted values and lower FEV_1_/FVC ratio, a significantly higher extent of desaturation, and a significantly lower 6MWD and DSP. Furthermore, the high-cadmium group had a significantly higher percentage of desaturation ≥4% after the 6MWT and post-test SpO_2_ of <90%.

### 3.4. Correlation between Blood Cadmium Levels and Desaturation during the 6MWT

Patients with COPD were stratified according to the quartiles of blood cadmium concentrations as follows: quartile 1 (<0.8 μg/L), quartile 2 (0.8–1.1μg/L), quartile 3 (1.1–1.5 μg/L), and quartile 4 (>1.5 μg/L). The frequency of desaturation (△SpO_2_ of ≥4% or △SpO_2_ of ≥4% and post-test SpO_2_ of <90%) after the 6MWT revealed a stepwise-increasing trend with an increase in quartiles of blood cadmium concentration (Figure 2).

### 3.5. Factors Associated with Desaturation during the 6MWT in Patients with COPD

After adjusting for significant confounding variables, a multivariable logistic regression model revealed that body weight and blood cadmium levels were independently associated with an increased risk of desaturation during the 6MWT, whereas pre-test SpO_2_ was independently associated with a decreased risk of desaturation during 6MWT (Table 3).

## 4. Discussion

The primary insight in this research was that blood cadmium levels, within the normal range, were independently associated with desaturation during the 6MWT in patients with COPD. Patients in the high-blood cadmium group had significantly worse pulmonary function tests (lower FEV_1_ % predicted, and lower FEV_1_/FVC ratio), a significantly higher risk of desaturation during the 6MWT, and significantly shorter 6MWD and DSP than those in the low-blood cadmium group.

The 6MWD is the primary outcome of the 6MWT and can be used to stratify patients with COPD to be included in studies to modify clinical outcomes [9,11,15], and a 6MWD less than 350 m significantly increases the risk of exacerbations, hospitalizations, and death [11]. Several studies also showed that the 6MWD was independently associated with the risk of death in COPD patients [10,12,13]. Our study demonstrated that the desaturation group had a lower 6MWD; however, the difference did not reach the significance level. The desaturation group had a significantly lower DSP than the no-desaturation group. Patients in the high-blood cadmium group had a significantly lower 6MWD and DSP than those in the low-blood cadmium group.

EID is associated with clinical outcomes and mortality in COPD [7,12,13,16]. The definition of desaturation during the 6MWT (i.e., EID) varies among previous studies, and is defined as a fall in SpO_2_ of ≥6% [12], fall in SpO_2_ of ≥4% or SpO_2_ of <90% [13], fall in SpO_2_ of ≥4% and SpO_2_ of <90% [7], or SpO_2_ of ≤88% [16] during the 6MWT. Some factors contribute to desaturation during the 6MWT. The Evaluation of COPD Longitudinally to Identify Predictive Surrogate Endpoints (ECLIPSE) study found that determinants of EID were obesity (BMI ≥ 30 kg/m^2^), impaired FEV_1_ (≤44% predicted), moderate or worse emphysema, and low SpO_2_ at rest (≤93%) [16]. COPD patients who experienced oxygen desaturation during the 6MWT had an increased risk of adverse outcomes than non-desaturation in COPD patients, including mortality [12], exacerbation frequency, declining lung function, and loss of lean body mass [7]. Our study revealed that 22% of patients exhibited EID, similar to the ECLIPSE study (21%); however, the definitions of EID were different. The desaturation group had significantly higher BMI, lower FEV_1_ predicted values, a higher percentage of GOLD stage III and IV, and lower SpO_2_ at rest. The above findings were also similar to the ECLIPSE study.

Tobacco smoking is the most important causal factor for developing COPD and a major non-occupational source of exposure to cadmium. Cadmium oxide, generated during tobacco smoking, may be deposited locally in lung tissue or absorbed into the systemic circulation. The absorption of cadmium in the lungs after smoking leads to blood concentrations that can be up to four or five times higher in tobacco smokers than nonsmokers [4]. An experimental study revealed that exposure to cadmium in soluble and nanoparticulate forms represented by cadmium chloride and cadmium oxide induced protein citrullination in cultured human lung epithelial cells [17]. Citrullination is a permanent post-translational protein modification, which is associated with autoimmune-mediated inflammatory responses. The immune system can attack citrullinated proteins, leading to autoimmune diseases, like in rheumatoid arthritis (RA), and found in COPD. The generation of anti-citrullinated protein antibodies (ACPAs), a hallmark of RA, might be an important factor in the development and pathophysiology of COPD [17,18].

Citrullination, autoimmunity, and persistent systemic inflammation had the roles on the development and courses of both COPD and RA [17]. Autoimmunity is likely to play a central role in the progression of COPD. Both antibody and cell-mediated responses appear to be involved in autoimmune responses and the development of lung damage in COPD [19]. A systematic review reported that COPD develops up to 68% more frequently in patients with RA, as compared to the general population [20]. A high number of specific ACPAs were associated with the presence of parenchymal lung abnormalities in patients with early, untreated RA [21]. However, smoking is a significant risk factor for both COPD and RA, and autoantibodies to modified self-antigens had also been described in COPD including ACPAs [22,23]. Whether RA had a causal effect on the development of COPD or the association resulted from a confounding factor of smoking—it is not well known. The possible impact of cadmium exposure inducing protein citrullination on the pathophysiology of COPD may require further studies, to include patients with COPD who also had RA.

Cadmium has high rates of soil-to-plant transfer and is a contaminant found in most human foodstuffs, which makes food a primary source of cadmium exposure in nonsmokers [5]. Currently, occupational exposure to cadmium is relatively less common among workers in developed countries due to regulatory activities [4]. However, a pooled analysis from two large longitudinal cohorts found that long-term occupational exposure to biological dust, mineral dust, and metals was associated with an accelerated decline in FEV_1_ and the FEV_1_/FVC ratio, which could potentially increase the risk of airway obstruction and COPD [24].

Cadmium exposure could induce oxidative stress, cell and tissue damage, and generate reactive oxygen species, activate apoptosis, induce endoplasmic reticulum stress and persistent chronic inflammation, disrupt extracellular matrix homeostasis, and contribute to post-translational modification of self-antigens (e.g., protein citrullination), and the formation of lymphoid follicles that contribute to the accumulation of autoreactive B and T cells, necessary for the development and persistence of autoimmune responses [19,25,26]. All of these alternations are potentially responsible for the pathogenesis of COPD, impaired lung function, and reduced gas exchange [4,5,27]. Thus, it is reasonable to assume that cadmium exposure may contribute to oxygen desaturation events during the 6MWT in patients with COPD.

Researchers have yet to determine the correlation between cadmium levels in human biofluid and EID in patients with COPD. A higher blood cadmium level, within the normal range, was associated with COPD prevalence in males, including those who had never smoked. Cadmium exposure, even in the low-dose range, is a risk factor for emphysema [28]. A large nationally representative sample of the US population demonstrated a significant association between serum cadmium concentration and obstructive lung disease. Active smokers with obstructive lung disease had significantly higher serum cadmium levels than normal control subjects, and a progressive increase in serum cadmium was associated with worsening lung function (FEV_1_ % predicted values) among smokers [29]. However, the above studies did not seek to determine the link between blood cadmium levels and EID in patients with COPD.

Our study showed that all patients in the high-blood cadmium group were smokers (current or former) and had significantly lower FEV_1_ (L), lower FEV_1_ % predicted, and a significantly lower FEV_1_/FVC ratio than those in the lower blood cadmium group. Furthermore, the high-blood cadmium group had a significantly higher percentage of desaturation during the 6MWT (i.e., EID), a significantly lower 6MWD, and a significantly lower DSP. A multivariable logistic regression model revealed that blood cadmium levels remained independently associated with EID. Although the correlation between blood cadmium levels and local deposition of cadmium in the lungs was unproved, blood cadmium concentrations could provide information on cadmium exposures [6] and are a feasible method in clinical practice to measure cadmium exposure. It is reasonable to speculate that cadmium deposition in the lungs could impair lung function and gas exchange and may influence exercise capacity and contribute to EID in patients with COPD.

The correlation between body weight and EID in COPD patients was not clearly determined. Weight loss is common in patients with COPD. However, more than one-third of patients with COPD are obese, and it may predispose them to increased risk for comorbidities [30,31]. Previous studies reported that obese patients with COPD had significantly reduced exercise capacity, as measured by the 6MWD, compared with non-obese patients with COPD, and there was no significant difference in desaturation during the 6MWT [32]. A multicenter prospective cohort study demonstrated that increasing obesity class was independently associated with worse respiratory-specific and general quality of life, reduced 6MWD, increased dyspnea, and greater odds of severe acute exacerbation [31]. Although the etiologies of exercise limitations in obese patients with COPD are multifactorial, and respiratory factors (a mixed obstructive-restrictive ventilatory deficit), increased metabolic loading, musculoskeletal abnormalities, and cardiocirculatory impairment may contribute [30], our study demonstrated that body weight was independently associated with EID.

A number of limitations hindered this study. First, this was a retrospective study conducted in a single tertiary-care referral facility, and the severity of COPD may be higher. Furthermore, the limited number of enrolled COPD patients (especially small numbers of GOLD stage IV patients) was noted. These facts no doubt limit the generalizability of our findings. Second, we only checked the baseline blood cadmium levels. We did not evaluate the degree of dyspnea and follow blood cadmium concentrations, pulmonary function tests, and exercise capacity during the 6MWT for a long period of time. The enrolled COPD patients were only followed up for one year, and the outcomes of acute exacerbation or mortality were not further investigated. Third, the major source of cadmium in COPD patients may be due to tobacco smoking in this study (91.8% of enrolled patients were former or current smokers); we did not investigate the possibility of other sources of cadmium exposure via occupational inhalation from cadmium-emitting industries, long-term dietary intake, air pollution, or dermal contact. We also did not assess the local deposition of cadmium in the lungs, which explicitly represents a tissue-specific accumulation of cadmium. Whether the blood concentrations of cadmium could accurately reflect the local accumulation of cadmium in the lungs is uncertain and requires verification from further studies. Finally, the exact causal relationship between blood cadmium levels and desaturation during the 6MWT may be difficult to determine due to the study’s retrospective nature, and there may be other potential risk factors or comorbidities of COPD that contribute to the results. The aim of this observational study was to evaluate the factors associated with desaturation during the 6MWT without considering issues of causality, and the residual or confounding variables that were not measured may influence results. Thus, our results should be interpreted carefully.

## 5. Conclusions

This study revealed that blood cadmium levels were independently associated with desaturation during the 6MWT in patients with COPD, and patients in the high-cadmium group had significantly worse lung function tests and lower 6MWD. Our results imply that efforts to avoid smoking, occupational exposure, and foods with high levels of cadmium may reduce EID events in patients with COPD. Further large-scale studies are required to verify the pathogenesis of cadmium-induced, smoking-related lung diseases, the cellular and molecular mechanism in lung diseases caused by cadmium exposure, and develop potentially prognostic tools or therapeutic agents targeting cadmium.

## Figures and Tables

**Figure 1 medicina-57-01160-f001:**
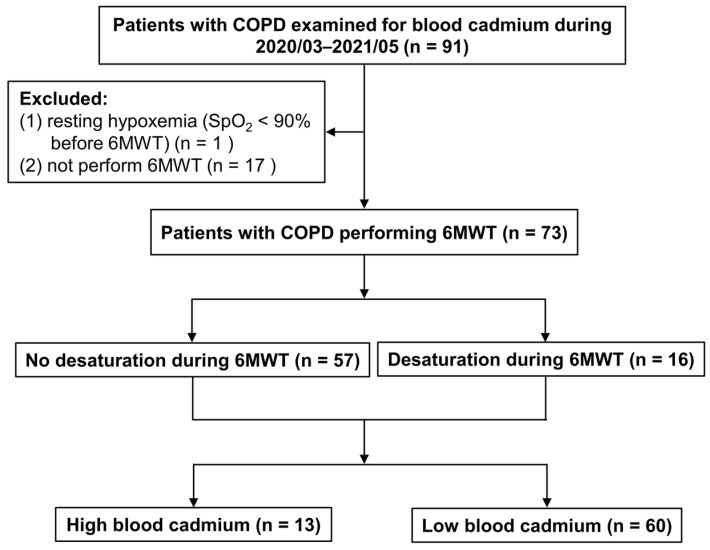
Flow chart of enrolled patients with COPD. COPD, chronic obstructive pulmonary disease; SpO_2_, arterial oxygen saturation measured by pulse oximetry; 6MWT, 6-min walk test.

**Figure 2 medicina-57-01160-f002:**
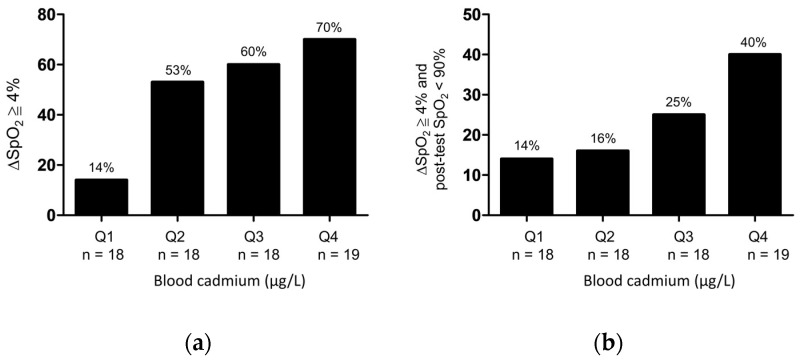
Frequency of (**a**) △SpO_2_ ≧ 4% or (**b**) △SpO_2_ ≧ 4%, and post-test SpO_2_ < 90% after 6MWT as stratified by quartiles of the blood cadmium concentrations in patients with COPD. Q1: first quartile (<0.8 μg/L); Q2: second quartile (0.8–1.1μg/L); Q3: third quartile (1.1–1.5 μg/L); Q4: fourth quartile (>1.5 μg/L).

**Table 1 medicina-57-01160-t001:** Background characteristics of COPD patients according to desaturation status during 6MWT.

Variables	Controls (n = 50)	COPD			*p*
		All (n = 73)	No-Desaturation (n = 57)	Desaturation (n = 16)	
Age (years)	53.6 ± 16.3	70.6 ± 9.1	71.0 ± 9.4	69.3 ± 8.1	0.491
Male (gender)		70 (95.9%)	55 (96.5%)	15 (93.8%)	0.530
Body weight (kg)		64.3 ± 12.7	61.8 ± 10.4	73.3 ± 16.4	0.030
Body mass index (kg/m^2^)		23.9 ± 4.2	23.1 ± 3.6	26.5 ± 4.9	0.009
Hypertension		28 (38.4%)	24 (42.1%)	4 (25%)	0.257
Diabetes mellitus		18 (24.7%)	13 (22.8%)	5 (31.3%)	0.489
Coronary artery disease		5 (6.8%)	4 (7%)	1 (6.3%)	1.000
Congestion heart failure		5 (6.8%)	4 (7%)	1 (6.3%)	1.000
Chronic liver disease		4 (5.5%)	2 (3.5%)	2 (12.5%)	0.207
Chronic kidney disease		6 (8.2%)	6 (10.5%)	0	0.328
Malignancies		10 (13.7%)	8 (14%)	2 (12.5%)	1.000
Osteoporosis		4 (5.5%)	3 (5.3%)	1 (6.3%)	1.000
Hemoglobin (g/dL)		14.0 ± 1.9	13.7 ± 2.0	15.3 ± 1.3	0.003
Serum creatinine (mg/dL)		1.0 ± 0.4	1.0 ± 0.3	1.0 ± 0.5	0.696
Blood cadmium (μg/L)	1.03 ± 0.56	1.29 ± 0.67	1.14 ± 0.46	1.81 ± 0.99	0.017
Smoking status					
Never		6 (8.2%)	4 (7.1%)	2 (12.5%)	0.606
Former		43 (58.9%)	34 (59.6%)	9 (56.3%)	0.807
Current		24 (32.9%)	19 (33.3%)	5 (31.2%)	0.875
FVC (L)		2.4 ± 0.7	2.5 ± 0.7	2.1 ± 0.6	0.057
FVC (%)		73.7 ± 18.9	76.3 ± 19.2	64.4 ± 14.6	0.026
FEV_1_ (L)		1.4 ± 0.5	1.4 ± 0.5	1.1 ± 0.3	0.003
FEV_1_ (%)		54.5 ± 17.7	57.7 ± 18.1	42.9 ± 9.5	<0.001
FEV_1_/FVC (%)		56.6 ± 9.8	57.5 ± 9.7	53.2 ± 9.8	0.121
GOLD classification					
Stage I		7 (9.6%)	7 (12.3%)	0	0.335
Stage II		32 (43.8%)	29 (50.9%)	3 (18.8%)	0.025
Stage III		32 (43.8%)	20 (35.1%)	12 (75%)	0.009
Stage IV		2 (2.8%)	1 (1.7%)	1 (6.2%)	0.393
Number of COPD exacerbations in the previous year					
0–1		7 (9.6%)	7 (12.3%)	0	0.335
≥2		4 (5.5%)	3 (5.3%)	1 (6.3%)	1.000
Six-minute walk test					
Pre-test saturation (%)		94.9 ± 2.7	95.6 ± 1.9	92.6 ± 3.8	<0.001
Post-test saturation (%)		90.9 ± 4.3	92.6 ± 2.3	84.8 ± 4.3	<0.001
Desaturation (%)		4.0 ± 3.1	2.9 ± 2.2	7.9 ± 3.1	<0.001
Pre-test heart rate (bpm)		86.4 ± 18.6	86.3 ± 19.4	86.6 ± 16.2	0.956
Post-test heart rate (bpm)		104.3 ± 13.6	103.7 ± 13.7	106.5 ± 13.6	0.466
Distance (m)		412.6 ± 99.1	416.2 ± 103.8	399.6 ± 82.2	0.558
Distance <350 m		14 (19.2%)	9 (15.8%)	5 (31.3%)	0.278
Distance saturation product (m%)		378.4 ± 86.0	390.1 ± 86.3	339.1 ± 74.3	0.036

Data are presented as numbers, mean and standard deviation, or median and interquartile range. The distance-saturation product is the product of the final walk distance in meters and the lowest oxygen saturation of the participants during 6MWT (as measured by pulse oximetry).

**Table 2 medicina-57-01160-t002:** Background characteristics of COPD patients as a function of blood cadmium levels.

Variables	High (>1.75 μg/L) n = 13	Low (≤1.75 μg/L) n = 60	*p*
Age (years)	68.3 ± 7.3	71.2 ± 9.4	0.310
Male (gender)	13 (100%)	57 (95%)	1.000
Body weight (kg)	66.3 ± 17.0	63.6 ± 11.8	0.629
Body mass index (kg/m^2^)	25.4 ± 6.8	23.6 ± 3.5	0.457
Hypertension	3 (23.1%)	25 (41.7%)	0.346
Diabetes mellitus	4 (30.8%)	14 (23.3%)	0.723
Coronary artery disease	0	5 (8.3%)	0.578
Congestive heart failure	1 (7.7%)	4 (6.7%)	1.000
Chronic liver disease	1 (7.7%)	3 (5%)	0.552
Chronic kidney disease	1 (7.7%)	5 (8.3%)	1.000
Malignancies	4 (30.8%)	6 (10%)	0.070
Osteoporosis	0	4 (6.7%)	1.000
Hemoglobin (g/dL)	14.7 ± 1.6	13.9 ± 2.0	0.182
Serum creatinine (mg/dL)	1.1 ± 0.5	1.0 ± 0.3	0.323
Blood cadmium (μg/L)	2.4 ± 0.6	1.0 ± 0.3	<0.001
Smoking status			
Never	0	6 (10%)	0.583
Former	7 (53.8%)	36 (60%)	0.683
Current	6 (46.2%)	18 (30%)	0.261
FVC (L)	2.1 ± 0.5	2.5 ± 0.7	0.110
FVC (%)	62.8 ± 15.7	76.0 ± 18.8	0.021
FEV_1_ (L)	1.0 ± 0.2	1.4 ± 0.5	<0.001
FEV_1_ (%)	44.5 ± 17.0	56.6 ± 17.2	0.023
FEV_1_/FVC (%)	51.1 ± 12.6	57.8 ± 8.8	0.025
GOLD classification			
Stage I	1 (7.7%)	6 (10%)	1.000
Stage II	2 (15.4%)	30 (50%)	0.031
Stage III	8 (61.5%)	23 (38.3%)	0.215
Stage IV	2 (15.4%)	1 (1.7%)	0.080
Number of COPD exacerbations in the previous year			
0–1	2 (15.4%)	5 (8.3%)	0.601
≥2	1 (7.7%)	3 (5%)	0.552
Six-minute walk test			
Pre-test saturation (%)	93.4 ± 4.6	95.3 ± 2.0	0.170
Post-test saturation (%)	87.5 ± 6.0	91.7 ± 3.5	0.029
Desaturation (%)	5.9 ± 3.5	3.6 ± 3.0	0.019
Desaturation ≥ 4%	10 (76.9%)	28 (46.7%)	0.048
Post-test saturation < 90%	7 (53.8%)	10 (16.7%)	0.004
Pre-test heart rate (bpm)	92.0 ± 10.9	85.2 ± 19.8	0.234
Post-test heart rate (bpm)	109.9 ± 7.6	103.1 ± 14.4	0.020
Distance (m)	378.8 ± 70.8	435.6 ± 87.4	0.033
Distance < 350 m	4 (30.8%)	10 (16.7%)	0.258
Distance saturation product (m%)	332.0 ± 70.4	395.9 ± 79.4	0.010

Data are presented as numbers, mean and standard deviation, or median and interquartile range. The distance-saturation product is the product of the final walk distance in meters and the lowest oxygen saturation of the participants during 6MWT (as measured by pulse oximetry).

**Table 3 medicina-57-01160-t003:** Multivariable logistic regression of factors associated with desaturation during 6MWT.

Factors	Odds Ratio (95% CI)	*p*
Univariate analysis		
Body weight (kg)	1.075 (1.018–1.135)	0.009
Body mass index (kg/m^2^)	1.225 (1.040–1.443)	0.015
Chronic liver disease	3.929 (0.508–30.392)	0.190
Hemoglobin (g/dL)	1.857 (1.186–2.906)	0.007
Blood cadmium concentration (μg/L)	4.188 (1.650–10.631)	0.003
FVC (%)	0.960 (0.925–0.996)	0.031
FEV_1_ (%)	0.934 (0.889–0.981)	0.006
FEV_1_/FVC (%)	0.014 (0.002–3.396)	0.127
Pre-test SpO_2_ (%)	0.588 (0.426–0.813)	0.001
Multivariate analysis		
Body weight (kg) (with each kg increase)	1.114 (1.002–1.239)	0.046
Blood cadmium concentration (μg/L) (with each μg/L increase)	12.849 (1.168–141.329)	0.037
Pre-test SpO_2_ (%) (with each unit (%) increase)	0.413 (0.192–0.890)	0.024

The multivariable analysis model included continuous variables (body weight, body mass index, hemoglobin, blood cadmium concentration, FVC (%), FEV_1_ (%), FEV_1_/FVC (%), and pre-test SpO_2_ of 6MWT) and a categorical variable (chronic liver disease). For the continuous variables, the odds ratio means that the risk of desaturation during 6MWT increases or decreases per unit increase of these variables.

## Data Availability

All data will be available from the corresponding author upon reasonable request.

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
