# Peer review of "Blood Cadmium Levels and Oxygen Desaturation during the 6-Minute Walk Test in Patients with Chronic Obstructive Pulmonary Disease"

_medicina, 2021, doi:10.3390/medicina57111160_

Round 1

Reviewer 1 Report

I have read the paper carefully and consider it very interesting because it indicates cadmium as a potential predictive risk factor for poor COPD outcomes. The article is well written, its methodology is clear and the discussion adequate so that it is easy to read and understand. The authors have clearly defined the limitations of their study. 

However, after another careful scrutiny, I do have one minor issue. Namely, in the Introduction, line 70, the authors explained that "The objective of the study was to determine the impact of blood cadmium levels…" I think that the word "impact" is not quite appropriate. The authors should consider using a more precise term e.g. "correlation" or similar.

Taking everything into consideration, I think that the authors provided an interesting topic and valuable paper for publishing.

Author Response

I have read the paper carefully and consider it very interesting because it indicates cadmium as a potential predictive risk factor for poor COPD outcomes. The article is well written, its methodology is clear and the discussion adequate so that it is easy to read and understand. The authors have clearly defined the limitations of their study. 

However, after another careful scrutiny, I do have one minor issue. Namely, in the Introduction, line 70, the authors explained that "The objective of the study was to determine the impact of blood cadmium levels…" I think that the word "impact" is not quite appropriate. The authors should consider using a more precise term e.g. "correlation" or similar.

Taking everything into consideration, I think that the authors provided an interesting topic and valuable paper for publishing.

Point 1: However, after another careful scrutiny, I do have one minor issue. Namely, in the Introduction, line 70, the authors explained that "The objective of the study was to determine the impact of blood cadmium levels…" I think that the word "impact" is not quite appropriate. The authors should consider using a more precise term e.g. "correlation" or similar.

Response 1: We thank the reviewer’s comment. We used the term “correlation” in the revised manuscript as follows (marked with red word):

The objective of this study was to determine the correlation between blood cadmium levels with desaturation during the 6MWT in patients with COPD.

We thank the reviewer for valuable comments. Addressing them fully has significantly strengthened the manuscript.

Reviewer 2 Report

Paper is interesting, well written, systematic. COPD is a widespread disease
and it would be more effective to have far more analyzed patients.
Also it is a very short analyzing period.
Together, these facts would result in better scientific confirmation.
Literature: 14/27 references is older than 5 years. You have to change it to bi smaller number

Author Response

Point 1: Paper is interesting, well written, systematic. COPD is a widespread disease
and it would be more effective to have far more analyzed patients. Also it is a very short analyzing period. Together, these facts would result in better scientific confirmation. 

Response 1: We thank the reviewer’s comment and agreed that more enrolled COPD patients followed up for a long period could result in better scientific confirmation.

  However, this is a retrospective study with the limited numbers of enrolled COPD patients (n = 73) due to only one-year enrolment period, and limit the generalizability of our findings

 We had described these above limitations in the last paragraph of the section of Discussion.

We will included more COPD patients and followed up for longer period in the future study.

Point 2: Literature: 14/27 references is older than 5 years. You have to change it to bi smaller number

Response 2: We thank the reviewer’s comment and apologized for the half of the references were older than 5 years. We had changed the lists of references and added twelve new references in the revised manuscript (all are not older than 5 years) (references 3, 9, 10, 15, 18, 19, 20, 21, 22, 23, 25, and 26). After change, 7/32 references is older than 5 years.

We thank the reviewer for valuable comments. Addressing them fully has significantly strengthened the manuscript.

Reviewer 3 Report

The article is devoted to an interesting subject and all experiments are described carefully and in detail. However, the authors must consider several issues, as follows:

  1. Line 26-27: I suggest reformulating the sentence: “Cadmium, a minor metallic element, is the major compound in tobacco smoke”, since the chemical composition of tobacco smoke consist of about more than 4000 compounds and Cadmium is only one of the main inorganic components among them.
  2. In the subchapter 1. Study Design and Patients Inclusion it is necessary to present in detail the two groups included in the study beyond the moderate extent presented in the Results section: the number of patients, the age range, the inclusion criteria and possible particularities of how to perform the test, if any. It is necessary to specify the fact that the test was done with room air, without supplementing the environment with oxygen.
  3. Line 87-88: Please reformulate the sentence “The distance-saturation product (DSP) is the product of the final walk distance in meters and the lowest oxygen saturation in room air during 6MWT [16]”, for a better understanding, with: “The distance-saturation product (DSP) is the product of the final walk distance in meters and the lowest oxygen saturation of the participants during 6MWT (as measured by pulse- oximetry [16]”.
  4. Line 96: please include “as an” to connect in the sentence “K2EDTA” with “anticoagulant”.
  5. Please explain in in a more detailed and organized manner the procedure for the inductively coupled plasma mass spectrometry analysis, for which ICP-MS acronym can be used (e.g. was the dilution 1 to 10 used to prepare your blood sample?, was the concentration of the nitric acid expressed as w/w?, was the range 0.8 to 40 μg/L used for both standards, etc.). Moreover, please clarify the sentence:” Values below the LOQ were assigned to LOQ for analysis.” since LOQ for cadmium was 0.5 μg/L.
  6. For Figure 1 and Tables 1-3 is not necessary to add explanations for acronyms which have already been presented before in the manuscript.
  7. Please include the explanation for what DSP stands for (Distance saturation product) in Tables 1 and 2.
  8. In subchapter 3. Comparing Patients with High and Low Blood Cadmium Levels the authors stated that: “….no significant differences were observed between the high blood cadmium and low blood cadmium group in terms of age, gender, body weight, body mass index, comorbidities, laboratory data except blood cadmium, and smoking status.”; which stands in contradiction with the p values for the smoking status for the two groups compared included in Table 2, which were >0.05. Authors must clarify this issue.
  9. In subchapter Discussion a more comprehensive debate on the connection between cadmium induced protein citrullination and the possible autoimmune responses which were identified in COPD smoking patients. Taking into account that anti-citrullinated protein antibodies (ACPAs) were observed in patients with rheumatoid arthritis (RA) and literature data showed that COPD develops up to 68% more frequently in patients with RA, as compared to the general population (e.g. Gergianaki et al, Journal of Comorbidity, 2019) it would have been interesting to include in the study groups patients with COPD who also had RA and to investigate the possible implications of cadmium in this direction in the pathophysiology of COPD.
  10. Line 264-265: Please add “exposure” at the end of the sentence “….to measure cadmium.”
  11. Multiple minor language corrections are necessary.
  12. A slightly more comprehensive bibliography would be welcome.

Author Response

The article is devoted to an interesting subject and all experiments are described carefully and in detail. However, the authors must consider several issues, as follows:

Point 1: Line 26-27: I suggest reformulating the sentence: “Cadmium, a minor metallic element, is the major compound in tobacco smoke”, since the chemical composition of tobacco smoke consist of about more than 4000 compounds and Cadmium is only one of the main inorganic components among them. 

Response 1: We thank the reviewer to point out this problem and we apologized for the lack of clarity. We revised the sentence as follows in the section of Abstract (marked with red text):

 Cadmium, a minor metallic element, is one of the main inorganic components in tobacco smoke.

Point 2: In the subchapter 1. Study Design and Patients Inclusion it is necessary to present in detail the two groups included in the study beyond the moderate extent presented in the Results section: the number of patients, the age range, the inclusion criteria and possible particularities of how to perform the test, if any. It is necessary to specify the fact that the test was done with room air, without supplementing the environment with oxygen.

Response 2: We thank the reviewer’s suggestions. We described the numbers of patients (173 COPD patients), the age range (40–78 years), the inclusion criteria and how to perform 6MWT. We also specify that 6MWT was done with room air, without supplementing the environment with oxygen.

  We added the above description in the subchapter 2.1. Study Design and Patients' Inclusion in the revised manuscript as follows (marked with red text):

  This study was based on retrospective analysis of 173 patients with COPD aged 40–78 years from the outpatient clinic or hospital admissions between March 2020 and May 2021 at Chang Gung Memorial Hospital (CGMH) in Taiwan. Inclusion criteria were as follows: (1) age > 40 years (2) post-bronchodilator FEV1/FVC < 0.7 (3) both blood cadmium examination and 6MWT were performed. COPD patients who performed 6MWT were under the supervision of well-trained technicians at the pulmonary rehabilitation center in our institution. The 6MWT was performed in accordance with the standard protocol [8], and the 6MWT was done with room air, without supplementing the environment with oxygen.

Point 3: Line 87-88: Please reformulate the sentence “The distance-saturation product (DSP) is the product of the final walk distance in meters and the lowest oxygen saturation in room air during 6MWT [16]”, for a better understanding, with: “The distance-saturation product (DSP) is the product of the final walk distance in meters and the lowest oxygen saturation of the participants during 6MWT (as measured by pulse- oximetry [16]”. 

Response 3: We thank the reviewer suggestion. We revised the sentence in the section of 2.2. Definitions as follows (marked with red text):

 The distance-saturation product (DSP) is the product of the final walk distance in meters and the lowest oxygen saturation of the participants during 6MWT (as measured by pulse- oximetry) [16]. 

Point 4: Line 96: please include “as an” to connect in the sentence “K2EDTA” with “anticoagulant”. 

Response 4: We thank the reviewer’s correction. We include “as an” to connect in the sentence “K2EDTA” with “anticoagulant” in the section of 2.4. Measurement of Blood Cadmium Levels in the revised manuscript 

Point 5: Please explain in in a more detailed and organized manner the procedure for the inductively coupled plasma mass spectrometry analysis, for which ICP-MS acronym can be used (e.g. was the dilution 1 to 10 used to prepare your blood sample?, was the concentration of the nitric acid expressed as w/w?, was the range 0.8 to 40 μg/L used for both standards, etc.). Moreover, please clarify the sentence:” Values below the LOQ were assigned to LOQ for analysis.” since LOQ for cadmium was 0.5 μg/L. 

Response 5: We thank the reviewer comments to describe the procedure for ICP-MS in a more detailed and organized manner. ICP-MS acronym was used. The blood sample were diluted 10 times with 1.5% (w/v) nitric acid. The concentration of the nitric acid was expressed as w/v (i.e., weight/volume). A standard calibration curve with a range of 0 to 40 μg /L was created.

 The lower limit of quantitation (LOQ) for cadmium was 0.5 μg/L. Since the LOQ was 0.5 ug/L, values below 0.5 μg/L were assigned to LOQ (i.e., 0.5 μg/L ) for analysis. For example, value of cadmium of 0.3 μg/L was considered as 0.5μg/L for analysis.

 We revised the above description of the procedure of ICP-MS in the section of 2.4. Measurement of Blood Cadmium Levels in the revised manuscript as follows (marked with red text):

  Blood cadmium measurements were conducted by inductively coupled plasma mass spectrometry (ICP-MS). Blood specimens were collected in 6 mL plastic blood collection tubes containing K2EDTA as an anticoagulant (BD, NJ, USA). Blood specimens were stored at 4 °C. Cadmium was quantified by means of ICP-MS on a PerkinElmer NexION 350X instrument (MA, USA) and analyzed using a no-gas mode. A 500 μL of blood specimens were diluted 10 times with 1.5% (w/v) nitric acid (JT Baker, NJ, USA) solution containing yttrium as an internal standard. The cadmium and yttrium standards were purchased from AccuStandard (CT, USA). A standard calibration curve with a range of 0 to 40 μg/L was created. The calibration curve had a correlation coefficient of more than 0.995. Level 1 control in-house prepared control and level 2 control Seronorm trace elements whole blood control (Sero, Billingstad, Norway) were used and analyzed at the start and end of each analytical run and after every 10 samples. The lower limit of quantitation (LOQ) for cadmium by ICP-MS was 0.5 μg/L. Since the LOQ for cadmium was 0.5 μg/L, the values below 0.5 μg/L were assigned to LOQ (i.e., 0.5 μg/L) for analysis. 

Point 6: For Figure 1 and Tables 1-3 is not necessary to add explanations for acronyms which have already been presented before in the manuscript. 

Response 6: We thank the reviewer’s suggestion. We deleted the explanations for acronyms which have already been presented before in the manuscript in Figure 1 and Tables 1-3. 

Point 7:  Please include the explanation for what DSP stands for (Distance saturation product) in Tables 1 and 2. 

Response 7: We thank the reviewer’s suggestion. We included the explanation for what DSP stands for (Distance saturation product) in the footer of Tables 1 and 2 as follows:

 The distance-saturation product is the product of the final walk distance in meters and the lowest oxygen saturation of the participants during 6MWT (as measured by pulse- oximetry). 

Point 8: In subchapter 3. Comparing Patients with High and Low Blood Cadmium Levels the authors stated that: “….no significant differences were observed between the high blood cadmium and low blood cadmium group in terms of age, gender, body weight, body mass index, comorbidities, laboratory data except blood cadmium, and smoking status.”; which stands in contradiction with the p values for the smoking status for the two groups compared included in Table 2, which were >0.05. Authors must clarify this issue. 

Response 8: We thank the reviewer to point out this problem and we apologized for the mistake. The smoking status between the high blood cadmium and low blood cadmium group was not significantly different (p > 0.05). We deleted the sentence “ and smoking status” in the revised manuscript. 

Point 9:  In subchapter Discussion a more comprehensive debate on the connection between cadmium induced protein citrullination and the possible autoimmune responses which were identified in COPD smoking patients. Taking into account that anti-citrullinated protein antibodies (ACPAs) were observed in patients with rheumatoid arthritis (RA) and literature data showed that COPD develops up to 68% more frequently in patients with RA, as compared to the general population (e.g. Gergianaki et al, Journal of Comorbidity, 2019) it would have been interesting to include in the study groups patients with COPD who also had RA and to investigate the possible implications of cadmium in this direction in the pathophysiology of COPD. 

Response 9: This is an excellent point of view. We appreciated the reviewer’s comments and suggestions to describe the connection between cadmium induced protein citrullination and the possible autoimmune responses in COPD patients who had smoking.

Citrullination is a permanent post-translational protein modification, associated with autoimmune-mediated inflammatory diseases. The immune system can attack citrullinated proteins, leading to autoimmune diseases, like in RA, and also found in COPD [1]. The generation of anti-citrullinated protein antibodies (ACPAs), a hallmark of RA and a break in immune tolerance, might be an important factor in the development of autoimmune diseases including COPD.

  Cigarette smoking results in numerous pathological processes, including oxidative stress, cell and tissue damage, chronic inflammation, post-translational modification of self-antigens (e.g. cadmium exposure induced protein citrullination in lung tissues), and the formation of lymphoid follicles that lead to the accumulation of autoreactive B and T cells necessary for the development and persistence of autoimmune responses [2].

Cadmium exposure contribute to protein citrullination and other post-translational modifications in lung tissues, and might contribute to the local production of ACPAs, which could lead to systemic autoimmunity. In other words, this mechanism of ACPA generation due to local lung inflammation might contribute to the development of RA [3].

 Citrullination, autoimmunity, and persistent systemic inflammation all had the roles on the development of both COPD and RA. A systematic review reported that COPD develops up to 68% more frequently in patients with RA, as compared to the general population [4]. However, the explanation of high incidence and causal relationship between COPD in RA patients still needed to be determined.

Cadmium exposure from smoking could induce protein citrullination, and patients with RA had ACPAs, which may contribute to the development and pathophysiology of COPD. However, autoantibodies to self-antigens had also been described in patients with COPD [4-6]. Smoking is the main risk factor of both development of COPD and RA. Therefore, whether RA had a causal effect on the development of COPD or the association was related to confounding factor of smoking is not well known.

We agreed with the reviewer that conducting studies to include patients with COPD who also had RA is interesting and could investigate the possible implications of cadmium in the pathophysiology of COPD. However, we apologized that we did not include patients who had COPD and RA due to the retrospective nature of our study. We will further include patients with rheumatoid arthritis and COPD in the future and investigate whether cadmium induced protein citrullination contribute to the pathophysiology of COPD by the presence of ACPAs.

 We added the above description in the fourth, fifth and seventh paragraphs of the Discussion section in the revised manuscript as follows (marked with red text):

  Citrullination is a permanent post-translational protein modification, which is associated with autoimmune-mediated inflammatory responses. The immune system can attack citrullinated proteins, leading to autoimmune diseases, like in rheumatoid arthritis (RA), and also found in COPD. The generation of anti-citrullinated protein antibodies (ACPAs), a hallmark of RA, might be an important factor in the development and pathophysiology of COPD [17, 18].

 Citrullination, autoimmunity, and persistent systemic inflammation had the roles on the development and courses of both COPD and RA [17]. Autoimmunity is likely to play a central role in the progression of COPD. Both antibody and cell-mediated responses appear to be involved in autoimmune responses and in the development of lung damage [19]. A systematic review reported that COPD develops up to 68% more frequently in patients with RA, as compared to the general population [20]. High number of specific ACPAs were associated with the presence of parenchymal lung abnormalities in patients with early, untreated RA [21]. However, smoking is a significant risk factor for both COPD and RA, and autoantibodies to modified self-antigens had also been described in COPD including ACPAs [22, 23]. Whether RA had a causal effect on the development of COPD or the association resulted from confounding factor of smoking is not well known. The possible impact of cadmium exposure induced protein citrullination on the pathophysiology of COPD may require further studies to include patients with COPD who also had RA to investigate.

 Cadmium exposure could induce oxidative stress, cell and tissue damage, and generate reactive oxygen species, activate apoptosis, induce endoplasmic reticulum stress and persistent chronic inflammation, disrupt extracellular matrix homeostasis, and contribute to post-translational modification of self-antigens (e.g. protein citrullination), and the formation of lymphoid follicles that contribute to the accumulation of autoreactive B and T cells necessary for the development and persistence of autoimmune responses [19, 25, 26]. 

References:

  1. Hutchinson, D.; Müller, J.; McCarthy, J.E.; Gun'ko, Y.K.; Verma, N.K.; Bi, X.; Di, Cristo. L; Kickham, L.; Movia, D.; Prina-Mello, A.; et al. Cadmium nanoparticles citrullinate cytokeratins within lung epithelial cells: cadmium as a potential cause of citrullination in chronic obstructive pulmonary disease. Int J Chron Obstruct Pulmon Dis. 2018, 13, 441-449.
  2. Pollard, K.M. Perspective: The Lung, Particles, Fibers, Nanomaterials, and Autoimmunity. Front Immunol. 2020, 11, 587136.
  3. Klareskog, L.; Catrina, A.I. Autoimmunity: lungs and citrullination. Nat Rev Rheumatol. 2015, 11, 261-2.
  4. Gergianaki, I.; Tsiligianni, I. Chronic obstructive pulmonary disease and rheumatic diseases: A systematic review on a neglected comorbidity. J Comorb. 2019, 9, 2235042X18820209.
  5. Byrne, R.; Todd, I.; Tighe, P.J.; Fairclough, L.C. Autoantibodies in chronic obstructive pulmonary disease: A systematic review. Immunol Lett. 2019, 214, 8-15. 
  6. Wen, L.; Krauss-Etschmann, S.; Petersen, F.; Yu, X. Autoantibodies in Chronic Obstructive Pulmonary Disease. Front Immunol. 2018, 9, 66. 

Point 10: Line 264-265: Please add “exposure” at the end of the sentence “….to measure cadmium.” 

Response 10: We thank the reviewer’s collection. We added the word “exposure” at the end of the sentence “….to measure cadmium.” in the revised manuscript. 

Point 11: Multiple minor language corrections are necessary. 

Response 11: We thank the reviewer to point out this problem. The manuscript has been edited by English editing service of MDPI (english-35837). 

Point 12: A slightly more comprehensive bibliography would be welcome. 

Response 12: We thank the reviewer’s suggestion. We added twelve new references in the revised manuscript (references 3, 9, 10, 15, 18, 19, 20, 21, 22, 23, 25, and 26). 

We thank the reviewer for valuable comments. Addressing them fully has significantly strengthened the manuscript.
